# Demographic Pattern and Hospitalization Outcomes of Depression among 2.1 Million Americans with Four Major Cancers in the United States

**DOI:** 10.3390/medsci6040093

**Published:** 2018-10-24

**Authors:** Rikinkumar S. Patel, Kuang-Yi Wen, Rashi Aggarwal

**Affiliations:** 1Department of Psychiatry, Griffin Memorial Hospital, Norman, OK 73071, USA; 2Cancer Prevention and Control Program, Fox Chase Cancer Center, Philadelphia, PA 19111, USA; Kuang-Yi.Wen@fccc.edu; 3Department of Psychiatry, Rutgers New Jersey Medical School, Newark, NJ 07101, USA; aggarwra@njms.rutgers.edu

**Keywords:** breast cancer, cancer, colorectal cancer, epidemiology, inpatient psychiatry, lung cancer, MDD, oncology, prostate cancer

## Abstract

Objective: To compare the prevalence of depression in the four most common cancers in the US and evaluate differences in demographics and hospital outcomes. Methods: This was a cross-sectional study using the Nationwide Inpatient Sample (2010–2014). We selected patients who had received ICD-9 codes of breast, lung, prostate, and colorectal cancers and major depressive disorder (MDD). Pearson’s chi-square test and independent sample *t*-test were used for categorical and continuous data, respectively. Results: MDD prevalence rate was highest in lung cancer (11.5%), followed by breast (10.3%), colorectal (8.1%), and prostate cancer (4.9%). Within colorectal and lung cancer groups, patients with MDD were significantly older (>80 years, *p* < 0.001) than non-MDD patients. Breast, lung, and colorectal cancer showed a higher proportion of female and Caucasian in the MDD group. Severe morbidity was seen in a greater proportion of the MDD group in all cancer types. The mean inpatient stay and cost were higher in the MDD compared to non-MDD group. Conclusion: Particular attention should be given to elderly, female, and to lung cancer patients with depression. Further studies of each cancer type are needed to expand our understanding of the different risk factors for depression as a higher proportion of patients had severe morbidity.

## 1. Introduction

Cancer is among the foremost causes of death worldwide. In 2012, there were 14 million newly diagnosed cancer cases and 8.2 million cancer-related deaths worldwide. The most common cancers in 2016 were breast cancer, lung and bronchus cancer, prostate cancer, and colon and rectum cancer [1]. Many cancer patients and survivors suffer from psychiatric problems, such as depression [2]. This may interfere with the patient’s capacity to cope with the burden of cancer, it may decrease utilization of treatment, extend hospitalization stay, decrease quality of life, and increase suicide risk [3,4,5]. Be that as it may, studies have shown overall prevalence rates of depression in the range of 0% and 58% amongst cancer patients [6]. To mention a few of the multitude of contributory factors that have resulted in this varied rate includes the use of different instruments to assess depression, utilization of several different diagnostic criteria to evaluate depression, and the variability of the cancer type and stage in the patient population [6,7].

From 2003 to 2012, breast cancer incidence rates were stable in Caucasian women and increased by 0.3% per year in African American women. Two systematic review studies concluded that patients with breast cancer are at higher risk of depression [8], which leads to increased functional impairment and poor treatment adherence [9]. Lung cancer is the second most commonly diagnosed cancer in both genders. From 2008 to 2012, lung cancer incidence rates reduced by 3.0% per year in men and by 1.9% per year in women. The association of depression with an adverse impact on patients’ quality of life [10], judgement on cancer treatment [4], care giver distress [11], and increased mortality [12] has been found among lung cancer patients.

The high prevalence of depression in colorectal cancer patients (23% to 44%) may be due to the severity of the disease itself as well as an additive factor of cancer post-treatment effects. For instance, other factors that have closely been associated with depression include advanced cancer stage, malaise, recurrence anxiety, pain, and chemotherapy [13,14,15]. Prostate cancer is very common in men and its risk is about 70% higher in African Americans than Whites. Depression has serious consequences for outcomes and recovery from prostate cancer as seen in incapacitating effects on health-related quality of life, functional status, health resource utilization, and hospitalization cost [16,17,18,19,20].

In the United States, reliable country wide data to determine the inpatient prevalence of depression in cancer patients are missing. Our study aims to compare the prevalence of depression in the four most common cancers in the United States and evaluate the differences in hospital outcomes in terms of morbidity, length of stay, and cost during hospitalization, using the largest nationwide inpatient sample (NIS) provided by the Agency for Healthcare Research and Quality (AHRQ).

## 2. Materials and Methods

### 2.1. Data Source

A retrospective analysis was performed using the Healthcare Cost and Utilization Project’s (HCUP) Nationwide Inpatient Sample (NIS) data from the years, 2010 to 2014 [21]. The HCUP-NIS database consists of the inpatient admissions of 4411 non-federal community hospitals and covering 45 states in the United States. About seven million records of inpatient admissions are recorded from these hospitals annually and we studied the inpatient data over the five-year period, 2010 to 2014 [21]. To represent all 50 states across the United States, we weighed the estimated samples using the discharge weights (DISCWT) variable available in the NIS data.

### 2.2. Selection Criteria

We identified adult patients above 18 years with a primary diagnosis of breast or lung or prostate or colorectal cancer using the International Classification of Diseases, Ninth Revision, Clinical Modification (ICD-9-CM) diagnosis codes. Then, based on the ICD-9-CM diagnosis codes, the patients with secondary diagnosis of MDD were identified as the target group and compared with cancer patients without MDD. In the NIS database, more than 14,000 ICD-9-CM diagnosis codes and 3900 procedure codes are further classified into clinically relevant categories by the AHRQ’s Clinical Classification Software (CCS) [22]. Using this feature of NIS, we were able to study a large population of relatively similar conditions into a single group. Breast cancer was identified using CCS code 24 (included ICD-9-CM codes, 174.0–174.6, 1748–175.0, 175.9, 233.0 or V10.3), lung cancer using CCS code 19 (included ICD-9-CM codes, 162.2–162.5, 162.8, 162.9, 209.21, 231.2 or V10.11), prostate cancer using CCS code 29 (included ICD-9-CM codes, 185, 233.4 or V10.46), colorectal cancer using CCS codes 14 and 15 (included ICD-9-CM codes, 153.0–153.9, 159.0, 209.10–209.16, 230.3 or V10.05), and MDD was identified using ICD-9-CM diagnosis codes, 296.2 and 296.3.

### 2.3. Variables of Interest

Our first goal was to measure the differences in demographic patterns in cancer patients between the MDD and non-MDD group, and the variables included age, gender, and race. Next, we evaluated the hospital outcome variables, including the severity of illness that measures the loss of body functions, inpatient length of stay and charges, and in-hospital mortality [23]. In the NIS, death during hospitalization is defined as in-hospital mortality. We calculated the inpatient length of stay as the number of nights the patient was hospitalized for the primary diagnosis (cancer types). Total charges of hospitalization per admission do not include professional fees and non-covered charges [23].

### 2.4. Approaches

The prevalence of depression in each cancer type over the five-year period was calculated using descriptive statistics. Cross-tabulation was used to summarize the differences in demographics and hospital outcomes between the MDD and non-MDD group. The mean and standard deviations (SD) were used to explain the continuous variables—age, length of stay, and cost—and were evaluated between both groups using an independent sample *t*-test. The *p* values for categorical data were generated using the Pearson’s chi-square test. We used discharge weight (DISCWT), which is given in the NIS database, to obtain national represent inpatient data. A *p* value < 0.001 was used as a reference to determine the statistical significance of the analysis. All statistical tests were done using SPSS version 23 (IRB Corp., Armonk, NY, USA) in this study [24].

### 2.5. Ethical Approval

Diagnostic and procedural information in the NIS was identified using the ICD-9-CM and CCS codes. To protect the privacy of patients, physicians, and hospitals, entire data was de-identified using key patient identifiers (KEY_ID). The use of administrative inpatient databases under the HCUP, according to the AHRQ of the U.S. The Department of Health and Human Services, does not require approval from an Institutional Review Board (IRB) as the NIS is a publicly available de-identified inpatient database.

## 3. Results

### 3.1. Sample Characteristics

Our study analyzed 2,121,020 patients diagnosed with cancer during inpatient admission from 2010 to 2014. Lung cancer (N = 675,579; 31.85%) was the most prevalent cancer type, followed by colorectal (N = 674,446; 31.8%), prostate (N = 394,620; 18.6%), and breast (N = 376,375; 17.75%) cancer. The patients’ with colorectal cancer were older (mean age, 68.49 ± 13.810 years) followed by lung cancer (67.98 ± 11.168 years), prostate cancer (63.28 ± 8.800 years), and those diagnosed with breast cancer were comparatively younger (59.12 ± 13.875 years). Male predominance was seen in prostate cancer (100%), lung cancer (51.6%), and colorectal cancer (51.1%).

### 3.2. Prevalence of Depression

The prevalence of MDD in the inpatient cancer population was highest in lung cancer (11.52%), followed by breast cancer (10.26%), colorectal cancer (8.09%), and lowest in prostate cancer (4.91%). Patients in the MDD group were comparatively younger than non-MDD patients, with breast cancer and lung cancer diagnoses (Table 1). The gender difference by presence of comorbid MDD is shown in Table 2. A statistically significant gender difference was seen in breast, lung, and colorectal cancer, which showed a higher female proportion in the MDD group than the non-MDD group.

Depression was most prevalent in breast cancer in the 41–60 (49.7%) age group, and, on the contrary, depression was prevalent in lung (58.8%), prostate (54.3%), and colorectal (46.7%) cancer in the 61–80 age group. In the elder population (>80 years), colorectal cancer (18.6%) and lung cancer (10.2%) showed the highest prevalence. MDD was prevalent in the youngest male age group (21–40 years) with breast cancer (6.6%) diagnosis.

A greater proportion of Whites were depressed in breast (80.5%), lung (84.5%), prostate (83.6%), and colorectal (81.6%) cancer. On the contrary, Blacks were less depressed when compared to the MDD and non-MDD group in breast cancer (9.5% vs. 14.7%), lung cancer (8.7% vs. 13.1%), prostate cancer (7.7% vs. 14.0%), and colorectal cancer (8.1% vs. 12.7%). In the Native Americans, all four types of cancer showed the lowest prevalence of MDD. A comparison of MDD prevalence (*p* < 0.001) by age groups and race is shown in Table 2.

### 3.3. Inpatient Outcomes

The mean length of inpatient stay was higher in the MDD group compared to the non-MDD group (Table 3). Inpatient hospitalization was highest among colorectal cancer patients (8.64 days vs. 7.74 days) followed by lung cancer (7.26 days vs. 6.63 days). Whereas, the mean length of inpatient stay was comparatively shorter in prostate cancer (2.59 days vs. 2.23 days) and breast cancer (2.89 days vs. 2.53 days). The mean inpatient charges were marginally higher in the MDD group compared to the non-MDD group in all cancer types except prostate cancer. Inpatient charges were highest among colorectal cancer patients (USD 71,714 vs. USD 69,948) followed by lung cancer (USD 63,621 vs. USD 61,626). Whereas, the mean inpatient charges were comparatively lower in the MDD group in prostate cancer (USD 42,949 vs. USD 43,038).

Severe morbidity was seen in a higher proportion of patients in the MDD group compared to the non-MDD group in all four cancer types. About 60.5% depressed lung cancer patients had severe morbidity, but the in-hospital mortality was lower in these patients compared to the non-MDD group (7.5% vs. 9.7%; *p* < 0.001). When 45.1% depressed colorectal cancer patients had severe disability due to loss of body function then there were lower reported inpatient deaths in them compared to the non-MDD group (2.6% vs. 3.3%; *p* < 0.001). A very low proportion of depressed breast cancer patients had severe morbidity (13.2%) and reported inpatient deaths (1.6%). Similarly, depressed prostate cancer patients had lowest reported inpatient deaths (0.9%) among all depressed cancer patients, though there were greater proportions with severe morbidity compared to the non-MDD group (14% vs. 9.6%; *p* < 0.001). The differences in morbidity and mortality between the MDD and non-MDD group across the four cancer types is shown in Table 4.

## 4. Discussion

This analysis showed depression was most common in lung cancer patients (11.52%) among the four most common cancer types in the United States. Depressed cancer patients displayed a marginally lower mean age and leaning toward a female predominance. Severe morbidity at admission was seen in a higher proportion of the MDD group in all cancer types. Due to this, the hospitalization length of stay and overall healthcare cost were higher in patients with comorbid MDD when contrasted with the non-MDD group.

Amongst the four most predominant cancer types in the US, the NIS dataset was used to compare the incidence of depression in the inpatient cancer population. Additionally, the effects of depression were also measured in accordance with hospitalization outcomes in these patients. From the results obtained, depression was most commonly seen at the highest five-year prevalence rate in patients with lung cancer. Moreover, the findings are consistent with studies involving a large population. From age 61 to 80, lung cancer ranks top on the list, in which depression was found to be the highest and predominantly observed in the female gender (63.8%). In a cross-sectional study from Scotland, about 13.1% lung cancer patients from an outpatient setting had comorbid depression [25]. Apart from this, the Korean national registry data study supported our results by their findings indicating the increased occurrence of psychiatric disorders, especially depression, in lung cancer patients [26]. Amongst all cancer types, lung cancer has been reported to have the highest psychological distress [27,28]. The result of grief in this group has been due to poor outcomes and the patient burden particularity during treatment [29].

Gender-specific cancers, such as breast and prostate cancers, showed a comparatively higher prevalence of depression in our current study. This pattern is the same as that seen in other studies. Jane et al., in a study of five selected cancer types, found that depression was most common in lung cancer (13.1%), followed by breast cancer (9.3%) [25]. The depression rate was the second most prevalent in breast cancer (10.26%), and was least common in prostate cancer (4.91%). In mid-aged breast cancer patients (41–60 yrs.), depression prevalence was relatively higher (49.7%) when contrasted with other cancers. This finding is in agreement with an earlier study, which showed that younger breast cancer survivors had a higher risk of depression [30]. Reports from an epidemiological study of breast cancer has shown that depressive disorder was diagnosed in 4.94% of women with breast cancer, which was higher than the over-all population [31].

Comparatively, in the general population, an increased rate of depression has been seen in females. In the MDD group, statistical significance was seen in colorectal, breast, and lung cancer for female predominance. This is similar with respect to various other studies with a mixed group of cancer types and various cancer stages [32]. In our study population, the MDD group in all cancer types was mainly consisting of an older mean age. Moreover, with further age-wise analysis, an increased MDD rate was found in the middle-aged group (41–60 yrs) and old-aged group (61–80 yrs). Results obtained from studies on cancer patients has shown older age to be a risk factor for psychiatric behavior [26]. More compelling evidence from other studies also shows the propensity of older cancer patients to have a higher risk of depression when an avoidance coping strategy is used [33]. The mental health services utilized by older patients may have an altered pattern when this patient population seek medical care and their views regarding the delivery of psychiatric diagnosis and treatment they are provided. With this in consideration, our current data does not take into cognizance the length of disease and variations that contribute to the individual differences leading to increased occurrences in older individuals, especially because these patients tend to have cancer in advanced stage, increased risk of recurrences, high susceptibility to physical symptoms, and longer treatment duration.

All cancer patients with depression had longer mean hospitalization duration compared to the non-MDD group in our study. A prospective study conducted by Nipp et al. concluded that the mean hospital stay of cancer patients was 6.3 days and factors associated with longer hospital stay were psychological distress (B = 0.11; *p* = 0.040) and depression symptoms (B = 0.22; *p* = 0.017) [34]. As per the cross-section study using Medical Expenditure Panel Survey (MEPS) data, the average annual health care expenditures was higher in depressed cancer patients ($18,401) compared to the non-depressed ones ($12,091) [35]. Inpatient expenditures in adult cancer survivors were 14% higher for depressed survivors than non-depressed survivors in the MEPS study and they proposed that depression may have a greater length of hospital stay or burden of illness, leading to increased expenditures [35]. Our study supports this fact, as the MDD group had higher hospitalization charges and longer hospitalization stays compared to non-MDD cancer patients, which could be seen as a higher proportion of cancer patients in the MDD group had severe morbidity at time of admission than that seen in the non-MDD group. Another retrospective study done at the University of California San Diego Healthcare System using administrative data concluded that depressed cancer patients had significantly more annual non-mental health provider visits (OR = 1.76), more likely to have an emergency department (ED) visit (OR = 2.45), overnight hospitalization (OR = 1.81), and 30-day hospital readmission (OR = 2.03) [36]. High rates of somatization, intensification of clinical symptom, and amplified responsiveness of bodily sensation are some of the possible reasons for the cancer patients with MDD to have increased healthcare service utilization. When compared with non-MDD patients, depressed patients are three to four times more likely to present with of nonspecific symptoms, such as fatigue, light headedness, headache, abdominal pain, and back pain [37]. Even cancer therapies may be associated with an increment in intensity of these symptoms, but the above two studies’ reports suggest that cancer patients with MDD may report recurrent and/or more extreme experiences of these non-specific symptoms, eventually leading to an increase of healthcare utilization [36,37,38].

There were few limitations in our study. This was an epidemiological study and therefore the accessible variables were limited. In our study, factors that were not considered include cancer trajectory, the length of disease, the stage of cancer at time of study, and previous medical history of any psychiatric disorders, which could alter the results significantly. The causal link between cancer and any psychiatric condition cannot be made due to the fact that our study design was cross-sectional, therefore the link is unclear. During the study period, few people may have been depressed before the final diagnosis, which could have resulted in an over-exclusion of cases. Moreover, a retrospective analysis that we carried could have been more valid and reliable if instead we had trained experts to carry out structured interviews. Furthermore, patients with MDD were only included in the study and therefore the true scope of depression could have been estimated. Last of all, we were unable to differentiate re-hospitalizations, and an accurate estimation of the total patient burden could not be made due to the nature of the administrative database. Nevertheless, our study has some strengths as we used the national sample data and included a large cancer inpatient population. With the selection criteria of the patients based on ICD-9 diagnosis codes in discharge diagnoses records, we reduced the selection bias. Studies in the past were done on small sample sizes and estimated a depression rate between 5% and 12%, whereas we used the nationwide US population and so our results are more generalized of the actual estimation of the prevalence then what our results may suggest. Additionally, the majority of the depression patients in our study were those during inpatient hospitalization for cancer treatment and this confirms the validity of their diagnoses.

## 5. Conclusions

Using the NIS data, we found that the prevalence rate of depression in cancer patients is in the range of 5% to 12%, which is higher than the general population. In addition, the prevalence of depression was highest in lung cancer patients among the four most common cancers in the United States. Our findings provide significant information for clinical concerns to diagnose and treat depression in cancer patients. Particular attention should be given to elderly, female, and to lung cancer patients with depression. Comorbid depression in cancer patients has a potential to drive hospital outcomes and costs, and early diagnosis for depression could have a significant impact on reducing healthcare utilization and costs. According to our literature review, this is the first nationwide inpatient study, including 4411 U.S. hospitals, to provide an estimate of depression prevalence in cancer patients, and the hospital outcomes, including length of inpatient stay and hospitalization cost, and in-hospital mortality associated with depression. The findings of our study recommend that future policy efforts are required to decrease excess healthcare expenditures related with depression in cancer patients. The results also highlight the importance of the integrated clinical care model in psycho-oncology to improve screening for depressive symptoms, preventing major depression, and appropriate management for depression.

## Figures and Tables

**Table 1 medsci-06-00093-t001:** Prevalence and the mean age difference between MDD and non-MDD groups.

Cancer	MDD Group		Non-MDD Group		*p* Value
	N	Mean Age ± SD	N	Mean Age ± SD	
Breast	35,017	58.86 ± 12.818	341,358	59.14 ± 13.979	<0.001
Lung	69,764	66.21 ± 10.895	605,815	68.18 ± 11.182	<0.001
Prostate	18,462	62.96 ± 8.805	376,158	63.30 ± 8.799	0.184
Colorectal	50,480	68.21 ± 13.667	623,966	68.51 ± 13.821	0.904

Chi-square test, *p* < 0.001. MDD: Major depressive disorder.

**Table 2 medsci-06-00093-t002:** Age group and race analysis between MDD and non-MDD subgroups.

Variable	Breast Cancer		Lung Cancer		Prostate Cancer		Colorectal Cancer	
	MDD (+)	MDD (−)	MDD (+)	MDD (−)	MDD (+)	MDD (−)	MDD (+)	MDD (−)
**Age, in years, in %**								
21–40	6.6	8.3	0.9	0.9	0.2	0.1	3.2	3.3
41–60	49.7	46.6	30.1	23.5	40.7	37.3	31.5	28.9
61–80	37.6	37.0	58.8	61.6	54.3	58.2	46.7	48.3
>80	6.1	8.1	10.2	14.0	4.8	4.3	18.6	19.5
**Gender, in %**								
Male	0.6	0.9	36.2	53.4	100.0	100.0	36.8	52.3
Female	99.4	99.1	63.8	46.6	0	0	63.2	47.7
**Race, in %**								
White	80.5	68.9	84.5	77.0	83.6	73.4	81.6	73.1
Black	9.5	14.7	8.7	13.1	7.7	14.0	8.1	12.7
Hispanic	5.9	8.8	3.7	4.7	4.7	7.1	6.5	7.8
Asian	1.2	4.0	0.7	2.7	0.6	1.8	1.3	3.3
NA	0.4	0.3	0.4	0.4	0.3	0.3	0.3	0.5
Other	2.5	3.3	2.0	2.3	3.1	3.4	2.3	2.7

Chi-square test, the results that were statistically significant with *p* < 0.001 are mentioned. MDD: Major depressive disorder; NA: Native American.

**Table 3 medsci-06-00093-t003:** Mean length of inpatient stay and inpatient charge differences between MDD and non-MDD groups.

Cancer	Mean Lenghth of Stay (±SD)			Mean Inpatient Charge (±SD)		
	MDD (+)	MDD (−)	*p* Value	MDD (+)	MDD (−)	*p* Value
Breast	2.9 (±3.34)	2.5 (±3.34)	<0.001	47,271 (±43157)	45,399 (±41,599)	<0.001
Lung	7.3 (±6.45)	6.6 (±6.15)	<0.001	63,621 (±67045)	61,626 (±69,419)	<0.001
Prostate	2.6 (±3.56)	2.2 (±2.89)	<0.001	42,949 (±32990)	43,038 (±34,695)	<0.001
Colorectal	8.6 (±8.96)	7.8 (±6.88)	<0.001	71,714 (±74671)	69,948 (±77,856)	<0.001

Independent sample *t*-test, the results that were statistically significant with *p* < 0.001 are mentioned. MDD: Major depressive disorder, SD: Standard deviation.

**Table 4 medsci-06-00093-t004:** Severe morbidity and In-hospital mortality analysis in MDD and non-MDD groups.

Variable	Breast Cancer		Lung Cancer		Prostate Cancer		Colorectal Cancer	
	MDD (+)	MDD (−)	MDD (+)	MDD (−)	MDD (+)	MDD (−)	MDD (+)	MDD (−)
Severe morbidity, in %	13.2	9.8	60.5	55.0	14.0	9.6	45.1	36.6
In-hospital mortality, in %	1.6	1.9	7.5	9.7	0.9	0.8	2.6	3.3

Chi-square test, the results that were statistically significant with *p* < 0.001 are mentioned. MDD: Major depressive disorder.

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
