# Peer review of "Demographic Pattern and Hospitalization Outcomes of Depression among 2.1 Million Americans with Four Major Cancers in the United States"

_medsci, 2018, doi:10.3390/medsci6040093_

Round 1
Reviewer 1 Report
This paper studied the prevalence of MDD in four types of cancer patients gathered from the HCUP-NIS. Although interesting, this paper needs some improvements:
Please report an estimate of how many cancer patients had MDD before cancer diagnosis (for example, 6, 12 and 24 months before observed diagnosis).
A unique subject identifier was used in the study. However, it is not clear if the patients were enumerated more than once in the computation of MDD.
It is not clear if the “primary diagnoses” were referred to absolute incident cases or to prevalent cases observed for the first time in the HCUP database.
In data sources, it was stated that the number of hospitals changed during 2010. Please describe how the five-year prevalence rate was computed for the period 2010-14.
In the limitations section, it was specified that an exclusion criterion was applied to MDD patients. Please report all inclusion/exclusion criteria in materials and methods section.
Table 2 (similarly to Table 1) should report the statistically significant comparisons (as yet explained in the text).
In Figures 1 and 2, please report graphically which comparisons were statistically significant.
In Figure 2, a chi-square test was used to compare groups. Please report graphically all the terms used in the 2x2 computation. Elseways, is it possible to substitute Figure 2 with a table.
Author Response
We are grateful for the review and the reviewer’s feedback has made the manuscript stronger. Thank you very much in advance for your consideration of this manuscript. Responses are mentioned below.
Please report an estimate of how many cancer patients had MDD before cancer diagnosis (for example, 6, 12 and 24 months before observed diagnosis) – This is one of the limitations of our study. We have elaborated this limitation in our limitation section (lines 310 onwards, paragraph before Conclusion section).
A unique subject identifier was used in the study. However, it is not clear if the patients were enumerated more than once in the computation of MDD – We cannot differentiate patients on readmissions. We have addressed this in our discussion more clearly as a limitation (line 327-329).
It is not clear if the “primary diagnoses” were referred to absolute incident cases or to prevalent cases observed for the first time in the HCUP database – These are the incident cases observed for first time, being a cross-sectional study.
In data sources, it was stated that the number of hospitals changed during 2010. Please describe how the five-year prevalence rate was computed for the period 2010-14 – The number of hospitals from where the data was retrieved remained persistent over the study period we looked into (4411 hospitals annual data from 2010 to 2014). It was calculated for the total period and have mentioned that descriptive analysis was done using SPSS (line 113-114).
In the limitations section, it was specified that an exclusion criterion was applied to MDD patients. Please report all inclusion/exclusion criteria in materials and methods section – Have added the selection criteria subsection in Methods to specify our study population.
Table 2 (similarly to Table 1) should report the statistically significant comparisons (as yet explained in the text). – All the values added were statistically significant. As per your recommendation we have mentioned the significant values in the footer. To ease the visibility for the readers, we arranged the table in as per the current design.
In Figures 1 and 2, please report graphically which comparisons were statistically significant. – Thank you. We have changed the figures 1 and 2 to Table 3 and 4 for better understanding.
In Figure 2, a chi-square test was used to compare groups. Please report graphically all the terms used in the 2x2 computation. Else ways, is it possible to substitute Figure 2 with a table – Thank you. We have changed the figures 1 and 2 to Table 3 and 4
Reviewer 2 Report
Although it is a retrospective cross-sectional study, it is targeted at 2.1 million Americans and has a certain significance. There is no particular comment on the contents.
I would like to point out some minor points.
On line 103, the space before "We" is too wide.
The Y axis should be included in Figure 1.
Indentation on line 165 is incorrect.
Author Response
We are grateful for the review and the reviewer’s feedback has made the manuscript stronger. Thank you very much in advance for your consideration of this manuscript. Responses are mentioned below.
I would like to point out some minor points. On line 103, space before "We" is too wide. The Y axis should be included in Figure 1. Indentation on line 165 is incorrect – Changes made as per the typo errors noted by the reviewer.
Round 2
Reviewer 1 Report
The paper has now been strongly improved and can be accepted for publication as it is.